# Reduced isometric knee extensor force following anodal transcranial direct current stimulation of the ipsilateral motor cortex

Ryan B. Savoury[1☯], Armin Kibele[2☯], Kevin E. Power[1☯], Nehara Herat[1☯], Shahab Alizadeh[1☯], David G. Behm[1☯]*

1 School of Human Kinetic and Recreation, Memorial University of Newfoundland, St. John's, Newfoundland and Labrador, Canada, 2 Institute for Sport and Sport Science, University of Kassel, Kassel, Germany

☯ These authors contributed equally to this work.
* dbehm@mun.ca

## Abstract

### Background

The goal of this study was to determine if 10-min of anodal transcranial direct current stimulation (a-tDCS) to the motor cortex (M1) is capable of modulating quadriceps isometric maximal voluntary contraction (MVC) force or fatigue endurance contralateral or ipsilateral to the stimulation site.

### Methods

In a randomized, cross-over design, 16 (8 females) individuals underwent two sessions of a-tDCS and two sham tDCS (s-tDCS) sessions targeting the left M1 (all participants were right limb dominant), with testing of either the left (ipsilateral) or right (contralateral) quadriceps. Knee extensor (KE) MVC force was recorded prior to and following the a-tDCS and s-tDCS protocols. Additionally, a repetitive MVC fatiguing protocol (12 MVCs with work-rest ratio of 5:10-s) was completed following each tDCS protocol.

### Results

There was a significant interaction effect for stimulation condition x leg tested x time [$F_{(1,60)}$ = 7.156, $p = 0.010$, $\eta p^2 = 0.11$], which revealed a significant absolute KE MVC force reduction in the contralateral leg following s-tDCS ($p < 0.001$, d = 1.2) and in the ipsilateral leg following a-tDCS ($p < 0.001$, d = 1.09). A significant interaction effect for condition x leg tested [$F_{(1,56)}$ = 8.12, $p = 0.006$, $\eta p^2 = 0.13$], showed a significantly lower ipsilateral quadriceps (to tDCS) relative MVC force with a-tDCS, versus s-tDCS [$t(15)$ = -3.07, $p = 0.016$, d = -0.77]. There was no significant difference between the relative contralateral quadriceps (to tDCS) MVC force for a-tDCS and s-tDCS. Although there was an overall significant [$F_{(1,56)}$ = 8.36, $p < 0.001$] 12.1% force decrease between the first and twelfth MVC repetitions, there were no significant main or interaction effects for fatigue index force.

**Data Availability Statement:** The datasets generated during and/or analyzed during the current study are publicly available from the Dryad

database (https://doi.org/10.5061/dryad.
brv15dvcn).

**Funding:** This research was partially funded by the
Natural Science and Engineering Research Council
of Canada David Behm: RGPIN-2017-0328.

## Conclusion

a-tDCS may be ineffective at increasing maximal force or endurance and instead may be
detrimental to quadriceps force production.

## Introduction

Transcranial direct current stimulation (tDCS) is a non-invasive brain stimulation technique
that can induce both excitatory and inhibitory cortical effects depending on the polarity of the
stimulation administered [1]. The effectiveness of tDCS for clinical use has shown positive
results involving the treatment of depression, anxiety, schizophrenia, Parkinson's disease,
chronic pain, stroke, and other neural-related problems [2–7]. However, the tDCS research on
athletic performance is not as consistent.

There is conflict in the literature as to whether tDCS can augment muscle strength and
endurance performance. There are many studies that have demonstrated that tDCS is effective
at increasing maximal muscle force and endurance [8–25] as well as strength training volume
[26]. Brief applications of anodal tDCS (a-tDCS) increased maximal voluntary force produc-
tion in both lower and upper limbs [10], pinch grip in stroke patients [27], and intramuscular
coherence in sustained low force hand muscle activity [28]. This augmented strength can be
sustained for 60 minutes after 20 minutes of tDCS over M1 [19] and has been attributed to an
enhancement of cortical motor drive to spinal motor pool [29].

However, many others report no significant effects or decreases of muscle force or endur-
ance when stimulating the motor cortex [9, 21, 28–37]. There were also no significant effects
on jump height [38], or single or repeated sprint performance [39, 40] when stimulating the
dorsolateral prefrontal cortex. Others have also reported that the significant decrease in maxi-
mal voluntary isometric force was accompanied by no significant differences in motor evoked
potentials (i.e., corticospinal excitability) [28] or voluntary muscle activation with t-DCS stim-
ulation targeting M1 [41]. Thus, in contrast to the aforementioned studies that show strength
increases, they indicate that cortical motor drive is not enhanced. This contradiction in the lit-
erature may be reflective of the diverse protocols utilized in these studies making comparisons
very difficult [42].

Meta-analyses of the overall literature tend to suggest small magnitude improvements in
muscle strength [43–46] and endurance [45–47]. The small magnitude benefits might have
been affected by low quality studies and selective publication bias [45]. Shyamali Kaushalya
et al. [47] in their meta-analysis on tDCS effects on cycling and running endurance showed a
positive effect on time to exhaustion, which they suggested may have resulted from increased
corticospinal excitability may influence ratings of perceived exertion. Increases in corticomo-
toneuronal excitability and decreases in short interval intracortical inhibition (SICI) were also
suggested to influence strength improvements in a tDCS training study [48].

Almost all tDCS studies provide stimulation contralateral to the tested muscle. A 2-week
strength training program with a-tDCS to the ipsilateral M1 provided prolonged (48 hours)
improvements in strength [48]. Only one study [19] has investigated the acute effect that tDCS
of the M1 can have on muscles ipsilateral to the site of stimulation reporting no changes in
knee extensors (KE) maximal force production. No studies to date have tested the acute effects
of a-tDCS on fatigue/endurance in muscles ipsilateral to the site of stimulation.

Examinations of cortical and spinal excitability of the contralateral, non-exercised limb
after an acute session of unilateral fatiguing exercise (monitoring of ipsilateral corticospinal
influences) have revealed conflicting results with both decreases [49, 50] and increases in

cortical excitability [51, 52] as well as increases [49, 50], decreases [51] and no significant change [52] in spinal excitability. Indeed Cabibel et al. [53] did discover cross facilitation or interhemispheric interactions with tDCS targeting the left M1 inducing both excitatory and inhibitory processing of the left M1. Similarly, a single session of strength training with a-tDCS increased strength, decreased SICI, and increased motor evoked potentials in the contralateral wrist extensor muscles [54]. In their review, Cabibel et al. [55] explained that tDCS-induced cross activation both reduces intracortical inhibition and increases interhemispheric excitatory inputs in the ipsilateral M1. This transfer could result in performance enhancements in muscles ipsilateral to the site of stimulation. Studies have also found evidence of interhemispheric facilitation of the motor cortices using sub-motor threshold intensity TMS stimulation, which suggests the existence of an underlying facilitatory neuronal circuit [56, 57]. One study using a single pulse supra-threshold TMS design demonstrated that a-tDCS could increase excitability of the contralateral motor cortex [58]. It is also possible that uncrossed corticospinal fibres that target ipsilateral motor neurons (10–30%) [59] and branched corticospinal fibres projecting to motor neurons bilaterally are affected. Although, this is less likely since these projections are strongest to axial muscles and may not be present for distal limb muscles [60].

A previous study reported that cathodal tDCS effects were greater in magnitude and duration for female participants when compared to males [61]. Similarly, a multivariate meta-regression revealed that women demonstrated greater magnitude responses to tDCS, which might be attributed to sex differences in the precise cortical anatomical locations, cognitive task strategies, as well as hormonal differences affecting brain stimulation [62]. Furthermore, the heterogeneity and genetic diversity of overall (both sexes) muscle strength, endurance, and corticospinal excitability findings is likely a result of variation in protocols [43].

The main goal of this study was to determine whether unihemispheric a-tDCS of the left M1 is capable of modulating maximal force production or fatiguability of either the contralateral or ipsilateral KE. It was hypothesized that there would be an increase in maximal force production and fatigue resistance in the contralateral and ipsilateral KE in relation to the site of tDCS. Due to the lack of literature on sex dependent effects of a-tDCS effects on motor function, this research question was exploratory.

## Materials and methods

### Participants

A priori power analyses (software package, G* Power 3.1.9.7: University of Dusseldorf, Germany) conducted using the results from studies by Hazime et al. [18] and Lattari et al. [14] with statistical power set at 0.8 and an effect size of 0.5 (moderate magnitude), suggested a required sample size of 10 and 8, respectively. Therefore, 16 healthy, participants were recruited for this study (8 males; age = 24.1 ± 2.8 years, height = 173.2 ± 8.3 cm, mass = 86.1 ± 13.3 kg and 8 females; age = 21.9 ± 1.6 years, height = 163.2 ± 8.6 cm, mass = 70.0 ± 14.7 kg). Participants were recreationally active with no history of musculoskeletal disorders and were screened for their suitability to receive tDCS based on tDCS checklist recommendations by Thair et al. [63], which included personal and family history of epilepsy, metal implants, implanted medication pump, pacemakers, recurring headaches, serious head injuries/surgeries, pregnancy, heart disease, and various medications. Participants were asked "which leg they would use to kick a ball at a target" to determine lower limb dominance [64]. All participants were determined to be right leg dominant. Each participant was required to read and sign a consent form and verbally consent to the researcher in order to participate in the study. This study was approved by the Interdisciplinary Committee on Ethics in Human Research at Memorial University of Newfoundland (ICEHR No. 20201316-HK) in accordance with the Declaration of Helsinki.

## Experimental design

This study utilized a fully randomized, crossover, repeated measures design, with all participants completing four protocols. The four protocols involved the participant receiving: 1) a-tDCS targeting the left M1, with testing of the contralateral (right) leg, 2) a-tDCS targeting the left M1 with testing of the ipsilateral (left) leg, 3) sham tDCS (s-tDCS) targeting the left M1, with testing of the contralateral (right) leg, and 4) s-tDCS targeting the left M1 with testing of the ipsilateral (left) leg. At least one week of recovery was allocated between each session to ensure that no effects from prior stimulations carried over to the next session (Fig 1) [63].

## tDCS intervention

A similar stimulation protocol as the only other study to test ipsilateral tDCS effects [19]. Using random allocation, participants underwent four sessions of tDCS (two a-tDCS and two s-tDCS) delivered via a direct current stimulator (TCT Research Limited, Hong Kong) using saline-soaked sponge electrodes. For all sessions, the anode (5 x 5cm) was placed at the left

---

**Warm-up**
Cycle ergometer at 70 rpm and 1 kilopond
3 isometric KE and KF contractions at 50% of perceived

---

**Pre-tests**
2-3 repetitions of 4-second MVCs with 1 minute recovery between repetitions

Ratings of perceived sensations (itching, tingling, scalp irritation)
on a scale from 1-10 (Likert scale: 1 = absent, 10 = severe)

---

| **Interventions** (10 -min durations of anodal-tDCS or sham-tDCS with minimum of 7 days between each protocol) | | | |
|---|---|---|---|
| Anodal-tDCS to the left M1, with testing of the contralateral (right) leg, | Anodal-tDCS to the left M1 with testing of the ipsilateral (left) leg | Sham-tDCS to the left M1, with testing of the contralateral (right) leg | Sham-tDCS to the left M1 with testing of the ipsilateral (left) leg |

---

**Post-test**
Ratings of perceived sensations (itching, tingling, scalp irritation)

Single KE MVC of either the left or right KE

Repeated contraction (fatigue) protocol consisting of 12 MVCs with a
work to rest ratio of 5:10 seconds

---

**Fig 1. Experimental design.** Abbreviations: KE: knee extensors, KF: knee flexors, MVC: maximal voluntary isometric contraction, tDCS: transcranial direct current stimulation.

M1, contralateral to the participant's dominant limb, with the cathode (5 x 7cm) placed on the shoulder area of the same side [8, 9, 28]. The M1 was located via the C3/C4 locations according to the 10–20 electroencephalography (EEG) electrode placement system [18–20, 29, 30]. a-tDCS protocols had a constant stimulation intensity of 2 milliamps (mA) with a duration of 10 minutes [21, 31, 33]. The s-tDCS protocols involved participants receiving 2 mA stimulation for the initial 30 seconds, followed by an additional 9.5 minutes of no stimulation [21]. Participants were not informed during the s-tDCS that the stimulator was not providing stimulation for the final 9.5 minutes. Prior research from our lab suggested that participants typically could not accurately differentiate between s-tDCS and a-tDCS protocols with 2mA stimulation. As described below, we also included a tDCS questionnaire to determine if the s-tDCS was an effective, blinding protocol.

## MVC and fatigue tests

Prior to any performance measurements, participants completed a five-minute warm up using a cycle ergometer at 70 revolutions per minute and 1 kilopond.

To measure force, a cuff with a non-extensible strap was attached to a strain gauge (Omega Engineering Inc., LCCA 500 pounds; sensitivity = 3 mV/V, OEI, Canada) and placed around the ankle of the participant. Knee joint angles were measured using a goniometer ($90^0$), since it has previously been shown that knee angle can affect isometric maximal voluntary contraction force (MVC) [65]. Before MVCs, participants were instructed to complete three warm-up contractions at what they perceived to be 50% of their maximum capability for five seconds each. Prior to each tDCS protocol, participants performed a minimum of two, four second MVCs for both the ipsilateral (left) and contralateral (right) KE. A third MVC was completed if the second MVC resulted in more than a five percent greater force than the initial contraction. One minute of recovery was provided between the MVCs. Participants were instructed to contract "as hard and as fast as possible", with consistent verbal encouragement being provided during the contractions [66]. Immediately following the final pre-test MVC, the intervention commenced. Immediately post-tDCS 10-minute protocol, participants performed a single KE MVC of either the ipsilateral or contralateral KE. Only a single MVC was performed to minimize the effect on the following fatigue protocol. Peak MVC forces were analyzed for the KE of the tested leg. All force data was sampled at 2000 Hz and analyzed with the software program (AcqKnowledge III, Biopac Systems Inc., Holliston, MA). Force was normalized by comparing the post-test tDCS MVC forces to pre-test tDCS values.

Following the post-tDCS MVC, participants performed a repeated contraction (fatigue) protocol consisting of 12 KE MVCs with a work to rest ratio of 5:10 seconds [67]. During this protocol, participants were not told how many contractions had been completed in order to minimize pacing effects [68–70]. All participants were similarly verbally exhorted to maximize each contraction [71], with a consistent form of encouragement, which involved stating "Go, Go, Go" once for each contraction. A fatigue index (FI: Eq 1) was calculated and analyzed.

$$FI = \frac{(Mean\ force\ of\ repetitions\ 1\&2) - (Mean\ force\ of\ repetitions\ 11\&12)}{Mean\ force\ of\ repetitions\ 1\&2}/100 \quad (1)$$

## tDCS blinding questionnaire

Before and after receiving tDCS (either a-tDCS or s-tDCS), participants were given a questionnaire, where they were asked to rate perceived sensations (e.g., itching, tingling, scalp irritation) on a scale from 1–10 (Likert scale: 1 = absent, 10 = severe) [63].

## Statistical analysis

Statistical analyses were calculated using SPSS software (Version 27.0, SPSS, Inc. Chicago, IL). Normality and homogeneity of variances tests were conducted for all dependent variables. If the assumption of sphericity was violated, the Greenhouse-Geiser correction was employed. For absolute MVC force, a three-way repeated measures ANOVA (2x2x2) with factors including time (pre-/post-tDCS), tDCS protocol (anodal/sham), and tested leg ((ipsilateral to tDCS) / contralateral to tDCS) was conducted. Sex was not included in the absolute data analysis as it is well established that men typically exert greater forces than women. However, for relative measures, sex was incorporated as a factor. For the fatigue index, a three-way repeated measures ANOVA (2x2x2) with factors including sex, tDCS protocol (anodal/sham), and leg tested was also completed. For the fatigue test, another 3-way repeated measures ANOVA (2x2x2) with factors including first and last (12) repetition, two legs and tDCS protocol (anodal/sham).

If significant interactions were detected, post-hoc t-tests corrected for multiple comparisons were conducted to determine differences between values. Significance was set at $p \leq 0.05$. Cohen's $d$ effect size was calculated to compare measures. Effect sizes were qualitatively interpreted as: (trivial <0.2, small $0.2 \leq d < 0.5$; medium $0.5 \leq d < 0.8$; large: $d \geq 0.8$) [72]. Day to day reliability for pre-test MVC was assessed with Cronbach's alpha intraclass correlation coefficient (ICC) [72].

Friedman's ANOVA was utilized to detect significant effects for scales related to headache, neck pain, blurred vision, scalp irritation, tingling, itching, burning sensation, acute mood change, fatigue, and anxiety. For significant effects post-hoc Wilcoxon Signed Ranks tests corrected for multiple comparisons were performed.

# Results

## Absolute force measures

Coefficient of variations (CV) less than 10% of the mean and excellent reliability described as an ICC: >0.9 were found between pre-test measurements for the contralateral ($\alpha$ = 0.943, a-tDCS CV: 31.6 = 6.5% of mean, s-tDCS CV: 30.5 = 6.3% of mean) and ipsilateral ($\alpha$ = 0.964, a-tDCS: 36.4 = 7.5% of mean, s-tDCS: 36.2 = 7.4% of mean) KE.

There was a significant [$F_{(1,60)}$ = 38.85, $p < 0.001$, $\eta p^2$ = 0.39] main effect for time with a small magnitude, 8.7% ($d$ = 0.27) decrease in KE MVC (both KE combined) from pre- to post-test, with no significant interaction effects found for condition (condition x time), or leg tested (leg tested x time), nor main effects for condition.

A significant [$F_{(1,60)}$ = 7.156, $p = 0.010$, $\eta p^2$ = 0.11] interaction effect was found for condition and leg tested (condition x leg tested x time) (Fig 2). Participants' maximal ipsilateral KE MVC force was reduced 14% (large effect size magnitude), following a-tDCS when compared to the pre-stimulation values [$t(15)$ = 4.35, $p < 0.001$, $d$ = 1.09], but no significant change for s-tDCS. There was also a significant [$t(15)$ = 4.84, $p < 0.001$, d = 1.21] MVC force reduction of 10.7% in the contralateral KE (contralateral to tDCS) following the s-tDCS protocol, but no significant difference following a-tDCS (Fig 2).

## Relative (normalized) MVC force

A significant interaction effect for condition x leg tested [$F_{(1,56)}$ = 8.12, $p = 0.006$, $\eta p^2$ = 0.13], showed a significantly lower ipsilateral quadriceps (ipsilateral to tDCS) relative MVC force with a-tDCS (85.92 ± 12.75%), versus s-tDCS (96.2 ± 10.5%) [$t(15)$ = -3.07, $p = 0.016$, $d$ = -0.77]. There was no significant difference between the relative contralateral quadriceps

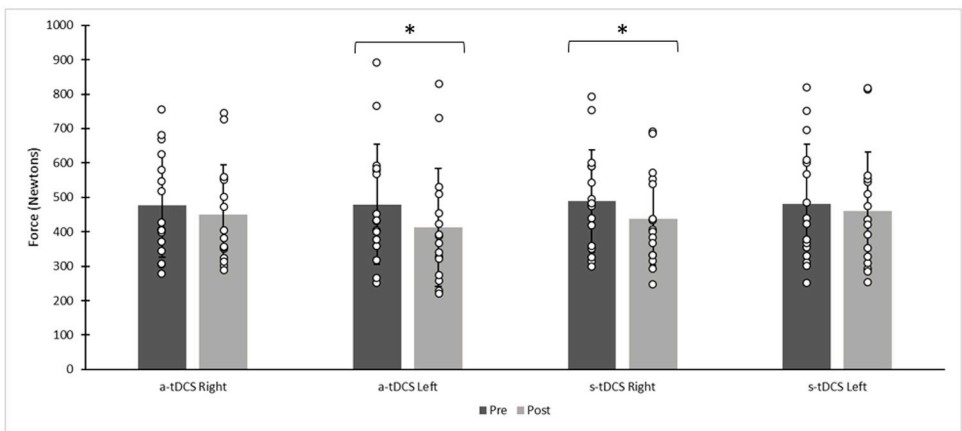

**Fig 2. Mean participant absolute contralateral and ipsilateral knee extensor force following anodal transcranial direct current stimulation (a-tDCS) or sham tDCS (s-tDCS).** (* denotes statistical significance at $p \leq 0.05$). Circles denote individual participant data and columns represent mean force values in Newtons.

(contralateral to tDCS) MVC force for a-tDCS and s-tDCS. No other significant main or interaction effects were observed. No significant sex effects were evident.

### Fatigue test

There were no significant main or interaction effects for fatigue index force (Fig 3). There was a significant main effect of the first and last repetitions with a moderate magnitude 12.05% (d = 0.54) decrease in force overall [$F_{(1,56)} = 8.36$, $p < 0.001$] (see Fig 4 for all 12 repetitions).

### tDCS blinding questionnaire data

Only minor side effects were reported following tDCS, including headache, scalp irritation, tingling, itching, burning sensation, and fatigue. Friedman's ANOVA revealed no significant differences in these variables between the four protocols (all p values $\geq 0.056$) suggesting that participants were sufficiently blinded to which type of stimulation they were receiving.

## Discussion

The main objective of this study was to determine if a-tDCS targeting the left M1 could modulate maximal force production or muscle fatiguability in either the right (contralateral to tDCS) or left (ipsilateral to tDCS) quadriceps. This study demonstrated significant force impairments (s-tDCS with testing of contralateral KE MVC, and a-tDCS with testing of ipsilateral KE MVC) and no significant changes (a-tDCS with testing of the contralateral KE MVC and s-tDCS with testing of the ipsilateral KE MVC) in discrete (single repetition) MVC tests.

Since four meta-analyses [43–46] report at least a small positive effect of tDCS on strength, we hypothesized an a-tDCS-induced increase in contralateral KE MVC force. However, this effect was not observed in the present study. The lack of significant change in normalized MVC force of the contralateral KE (contralateral to tDCS) following a-tDCS, in comparison to s-tDCS was contradictory to many studies which have reported force increases following anodal stimulation of the M1 [10–12, 18, 19]. Although only one of the studies that reported increases tested the KE [19], a larger number of studies testing the KE following a-tDCS reported no significant changes in comparison to the control [29, 30, 32, 73]. Additionally,

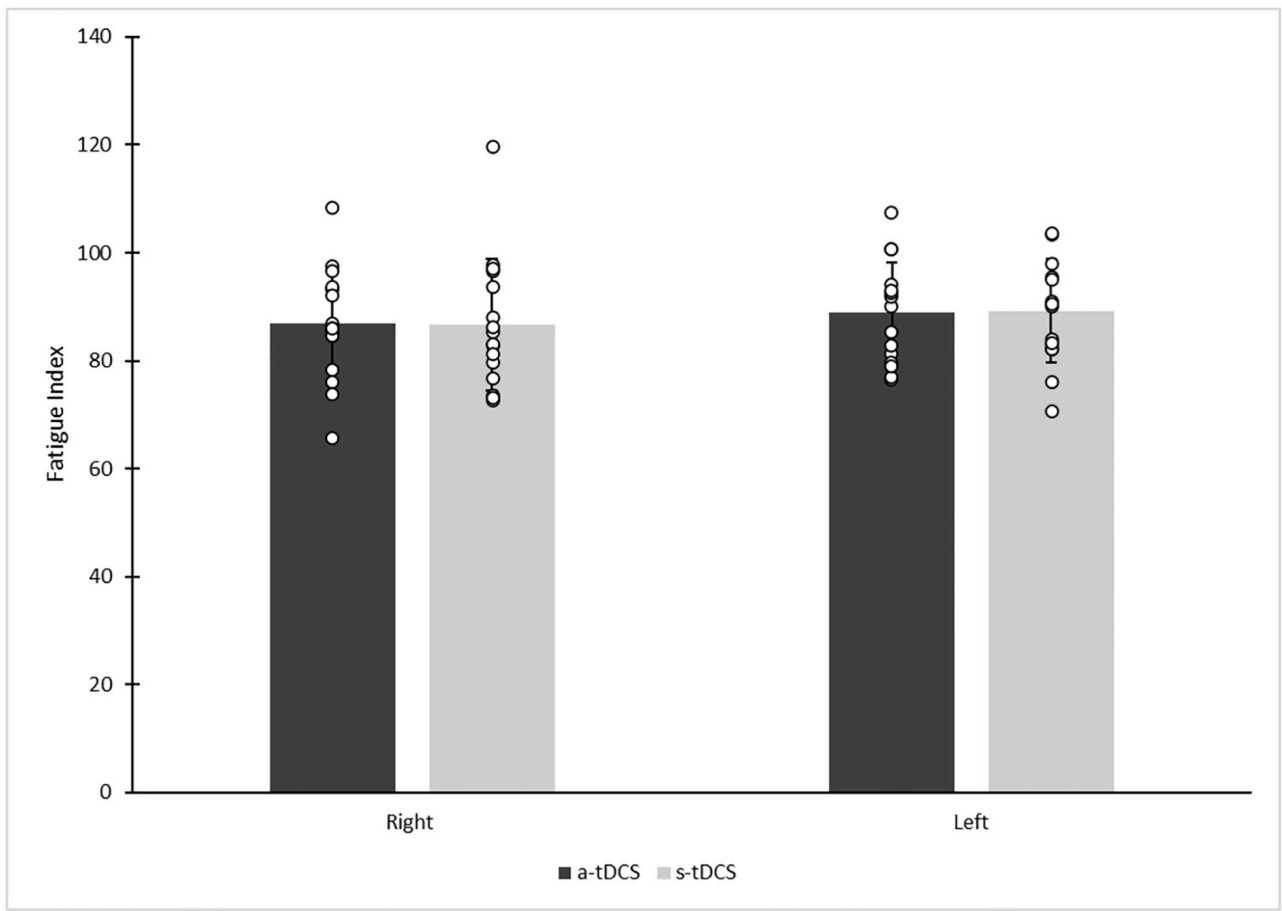

**Fig 3. Participant contralateral and ipsilateral knee extensor force fatigue index following anodal transcranial direct current stimulation (a-tDCS) or sham tDCS (s-tDCS).** (* denotes statistical significance at $p \leq 0.05$). Circles denote individual participant data and columns represent mean force values in Newtons.

following a-tDCS in the present study, 4/16 (2 males, 2 females) and 2/16 (2 females) participants experienced increased MVC force in their contralateral and ipsilateral quadriceps, respectively. This inter-individual variability likely contributed to some of the non-significant results of this study, and in combination with previous research suggests that a-tDCS is not a consistently effective ergogenic aid when the goal is to increase maximal KE force for a discrete contraction. This lack of reliability and high variability in the literature may be also related to the great diversity of implemented protocols (e.g., differences in electrode location, size, number, current density, polarity, and stimulation duration) [42].

The anodal and sham tDCS post-test protocols were conducted after approximately 10 minutes of physical inactivity. While the pre-test MVCs were performed shortly after a warm-up, the beneficial effects of this warm-up may have been reduced after 10 minutes of inactivity [74, 75]. The reported significant force losses with contralateral s-tDCS, and ipsilateral a-tDCS might be attributed to a diminished warm-up-induced post-activation potentiation enhancement that can increase force through phosphorylation of myosin light chains, and increased muscle temperature [76]. However, a previous studies involving a-tDCS of the contralateral leg motor cortex reported improved foot pinch force for 30 minutes post-a-tDCS [10], as well as increased KE MVC force for 60 minutes after a-tDCS [19]. Similarly, the positive effects of

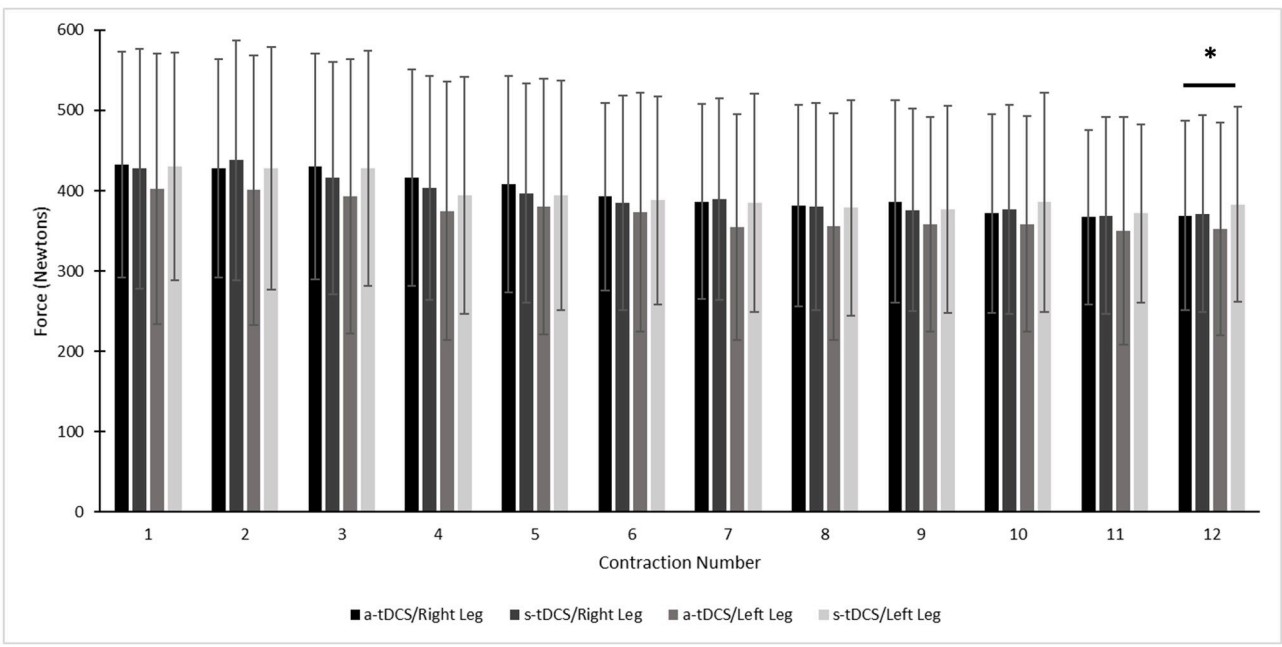

**Fig 4. Mean participant contralateral and ipsilateral knee extensor (KE) force for each contraction of the fatigue protocol.** Asterisk highlights a significant (p<0.0001) decrease in force from repetition #1 to #12. Columns and bars represent mean force values and standard deviations in Newtons.

warm-up activities on subsequent performance have been reported to be sustained for 8–12 min [77, 78], 10–15 min [79], 18 minutes (with adolescents) [80], and 20 minutes [81, 82].

While a-tDCS in the literature has shown the ability to increase M1 excitability, it has also been hypothesized that a-tDCS could attenuate the reduction in output from the M1 contributing to supraspinal fatigue [21, 83]. While there were no significant fatigue index interactions between legs or stimulation conditions, there was a significant overall (main effect for contractions) 12.05% force decrease between the first and last contractions of the protocol (Fig 4). It has previously been demonstrated that a-tDCS can delay the onset of fatigue for a prolonged submaximal contraction [9, 21, 24, 25], although again, only one of these studies tested the KE [21]. Numerous other studies also reported no significant changes in KE fatiguability following a-tDCS [21, 30, 31] with one study also reporting increased muscle fatiguability [32]. While Angius et al. [21] suggested an extracephalic electrode montage was more effective at inducing a-tDCS effects on muscle fatigue than cephalic montages, our study found that an extracephalic montage was ineffective in modulating muscle fatigue. It is possible that the difference in the fatigue test used may have led to this discrepancy, with our study utilizing a 12 x 5s MVC protocol while Angius et al. [21] utilized a submaximal 20% MVC force time to exhaustion test. Furthermore, Shyamali Kaushalya et al. [47], only demonstrated tDCS induced effects with a time to exhaustion test during running and cycling endurance activities. Additionally, anodal tDCS did not influence performance during repeated "all out" efforts (4–6 sec) interspersed with 30 sec rests [39]. Although the fatigue protocol in the present study involved 12 5-s MVCs with 10-s rest, the accumulated fatigue only reduced the post-test MVC by a moderate magnitude 12.05% (d = 0.54). This suggests that a-tDCS is more effective for delaying the effect of muscle fatigue for low intensity activities, while those requiring maximal exertions may not experience similar benefits.

This study found no significant differences between male and female participants for relative force production or fatigue index. This suggests that tDCS affects male and female

participants to a similar extent. A review by Dedoncker [62] suggested that women demonstrated greater accuracy on cognitive tasks with a-tDCS. A previous study that retrospectively re-analyzed data collected from previous transcranial direct current stimulation studies did not report significant cortical excitability differences between the sexes for a-tDCS. However, the effects of cathodal tDCS on neuroplasticity were greater, lasted longer and demonstrated more inhibition for female participants in comparison to males, suggesting that female participants may experience increased effects of tDCS possibly due to the effects of sex hormones [61]. These sex differences might be attributed to differences in identifying cortical anatomical locations, sex-dependent cognitive task strategies, and hormonal differences affecting brain stimulation. Our findings suggest that a-tDCS had similar performance effects for both male and female participants. More research is needed to explore possible sex differences.

## Limitations

It might be suggested that the recruitment of 16 participants in this study involving a more variable intervention might have led to low statistical power even though the power analysis indicated that 8–10 participants should provide sufficient power. This lower power may have diminished the possibility of revealing any sex differences. Furthermore, as only the KE (quadriceps) were tested, future studies should aim to determine if other muscle groups ipsilateral to the site of stimulation are affected in a similar manner, while also utilizing transcranial magnetic stimulation (TMS) to determine potential changes in corticospinal excitability. When combined with transmastoid electrical stimulation which activates axons in the spinal cord, effects can be distinguished between cortical and spinal excitability or inhibition. Finally, as reviewed by Savoury et al. [42], differences in stimulation protocols (e.g., duration, intensity, electrode location, blinding efficacy, individual expectations and others) make it difficult to make direct comparisons with other studies. A series of comprehensive studies are needed to determine optimal stimulation protocols suitable for various populations (e.g., possible differences between sexes, age), specific performance enhancement (e.g., strength, endurance, or cognitive) and more research targeting areas other than the M1 (e.g., temporal cortex and prefrontal cortex).

## Conclusions

This study found that 10 minutes of 2 mA of a-tDCS is not an effective or consistent method for increasing maximal force production or reducing fatigue in the KE either contralateral or ipsilateral to the stimulated M1. Furthermore, the decrements with the contralateral s-tDCS also contributed to the inconsistent findings with tDCS. With many athletes looking to devices such as those for administering tDCS to provide performance enhancements, it is important to caution that tDCS may not be beneficial but could instead be detrimental to exercise performance.

## Acknowledgments

The experiments comply with the current laws of the country in which they were performed.

## Author Contributions

**Conceptualization:** Ryan B. Savoury, Armin Kibele, Kevin E. Power, Shahab Alizadeh, David G. Behm.

**Data curation:** Ryan B. Savoury, Nehara Herat, Shahab Alizadeh.

**Formal analysis:** Ryan B. Savoury, Shahab Alizadeh, David G. Behm.

**Funding acquisition:** David G. Behm.

**Investigation:** Ryan B. Savoury, David G. Behm.

**Methodology:** Ryan B. Savoury, Armin Kibele, Nehara Herat, Shahab Alizadeh, David G. Behm.

**Project administration:** Ryan B. Savoury, David G. Behm.

**Resources:** Ryan B. Savoury.

**Supervision:** David G. Behm.

**Validation:** Kevin E. Power.

**Writing – original draft:** Ryan B. Savoury.

**Writing – review & editing:** Armin Kibele, Kevin E. Power, Nehara Herat, Shahab Alizadeh, David G. Behm.

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
