## [Decision Letter · Decision Letter 0]

12 Jul 2022

PONE-D-22-11905Reduced Isometric Knee Extensor Force Following Anodal Transcranial Direct Current Stimulation of the Ipsilateral Motor CortexPLOS ONE

Dear Dr. Behm,

Thank you for submitting your manuscript to PLOS ONE. After careful consideration, we feel that it has merit but does not fully meet PLOS ONE’s publication criteria as it currently stands. Therefore, we invite you to submit a revised version of the manuscript that addresses the points raised during the review process.

We look forward to receiving your revised manuscript.

Kind regards,

Xin Ye, Ph.D.

Academic Editor

PLOS ONE

Journal Requirements:

“This research was partially funded by the Natural Science and Engineering Research Council of Canada (Grant number 2017-03728). The experiments comply with the current laws of the country in which they were performed. The authors have no conflict of interest to declare. The datasets generated during and/or analyzed during the current study are not publicly available, but are available from the corresponding author who was an organizer of the study.”

“This research was partially funded by the Natural Science and Engineering Research Council of Canada

David Behm: RGPIN-2017-0328”

4. We note you have included a table to which you do not refer in the text of your manuscript. Please ensure that you refer to Table 2 in your text; if accepted, production will need this reference to link the reader to the Table.

Reviewers' comments:

Reviewer's Responses to Questions

**Comments to the Author**

1. Is the manuscript technically sound, and do the data support the conclusions?

Reviewer #1: Yes

Reviewer #2: No

Reviewer #3: Partly

2. Has the statistical analysis been performed appropriately and rigorously? 

Reviewer #1: Yes

Reviewer #2: Yes

Reviewer #3: I Don't Know

3. Have the authors made all data underlying the findings in their manuscript fully available?

Reviewer #1: Yes

Reviewer #2: No

Reviewer #3: Yes

4. Is the manuscript presented in an intelligible fashion and written in standard English?

Reviewer #1: Yes

Reviewer #2: Yes

Reviewer #3: Yes

5. Review Comments to the Author

Reviewer #1: The manuscript is a technically sound piece of scientific research with data that supports the conclusions. It appears that all appropriate statistical analyses were utilized to reach the authors' conclusions.

Reviewer #2: The aim of this study was to determine if anodal transcranial direct current stimulation (a-tDCS) to the motor cortex (M1) modulates quadriceps isometric maximal voluntary contraction (MVC) force or fatigue contralateral or ipsilateral to the stimulation site. Authors main conclusions are that a-tDCS may be ineffective at increasing maximal force or endurance and instead may be detrimental to quadriceps force production.

Because of the rise in popularity of a-tDCS as a tool with potential to influence sport performance, the aims of the present study could be considered relevant to the field. However, the manuscript has some limitations that decreased the initial enthusiasm.

First of all, a single blind approach (with only subjects blinded to condition) could not be the best way to perform this research. Double blind is recommended in this kind of study in which investigators can inadvertently influence the experiment’s results due to hypothesis expectations (i.e., for example, with differences in the way subjects are encouraged during the session).

Another question, is the protocol used to induce KE fatigue, which from my point of view does not work well (induced only a 12% decrease in force overall). A time to exhaustion test (TTE) would have been desirable, or at least, a large number of repetitions to increase the perception of effort to a greater extent, since it has previously been argued that modulation of RPE could underpin the tDCS induced beneficial effects on endurance performance.

The other main weaknesses of the study are related to the way it is written. There is a lack of explanations of arguments to support a priori hypothesis and a rather superficial discussion about the results. There are also some inconsistencies between the statistical analysis done and the way results are interpreted and the conclusions raised does not align well with results.

Introduction

Introduction is confusing and the rationale is not clear at all. Overall, introduction needs reorganization of content and a better justification of the aims of the study.

For example, there is no proper explanation of the physiological rationale for the effects of a-tdcs over M1 on muscle force. All the justification is based on the effects of a-tdcs on MEP amplitude. Although elucidating the mechanisms of a-tdcs is not the aim of the present study, a proper justification of the selection of stimulating areas, explaining possible physiological mechanisms by which they might influence motor behavior, should be included. Furthermore, it should be better physiologically justified the interaction between both motor cortices and its reciprocal influences, both in terms of facilitation and inhibition, when altering the “excitability” of one hemisphere. The rationale of cross education does not fit well in the intro, or at least at the main argument.

Lines 137 – 140: This is too speculative Hard to believe that tDCS may exert an effect on the reticulospinal system. My recommendation is to delete this sentence.

There is a need to actualize the citing literature, because there is a lot of missing references regarding the effects of TDCS on maximal force production and endurance performance. See some of them:

The Effect of Anodal Transcranial Direct Current Stimulation on Quadriceps Maximal Voluntary Contraction, Corticospinal Excitability, and Voluntary Activation Levels.

Kristiansen M, Thomsen MJ, Nørgaard J, Aaes J, Knudsen D, Voigt M.

J Strength Cond Res. 2021 Mar 3. doi: 10.1519/JSC.0000000000003710. Online ahead of print.

Anodal transcranial direct current stimulation enhances strength training volume but not the force-velocity profile.

Alix-Fages C, García-Ramos A, Calderón-Nadal G, Colomer-Poveda D, Romero-Arenas S, Fernández-Del-Olmo M, Márquez G.

Eur J Appl Physiol. 2020 Aug;120(8):1881-1891. doi: 10.1007/s00421-020-04417-2.

Transcranial Direct Current Stimulation Does Not Improve Countermovement Jump Performance in Young Healthy Men.

Romero-Arenas S, Calderón-Nadal G, Alix-Fages C, Jerez-Martínez A, Colomer-Poveda D, Márquez G.

J Strength Cond Res. 2019 Jul 31. doi: 10.1519/JSC.0000000000003242. Online ahead of print.

Transcranial direct current stimulation and repeated sprint ability: No effect on sprint performance or ratings of perceived exertion.

Alix-Fages C, Romero-Arenas S, Calderón-Nadal G, Jerez-Martínez A, Pareja-Blanco F, Colomer-Poveda D, Márquez G, Garcia-Ramos A.

Eur J Sport Sci. 2021 Feb 25:1-10. doi: 10.1080/17461391.2021.1883124. Online ahead of print.

Acute effects of transcranial direct current stimulation on cycling and running performance. A systematic review and meta-analysis.

Shyamali Kaushalya F, Romero-Arenas S, García-Ramos A, Colomer-Poveda D, Marquez G.

Eur J Sport Sci. 2021 Jan 7:1-13. doi: 10.1080/17461391.2020.1856933. Online ahead of print.

Effects of Anodal Transcranial Direct Current Stimulation on Training Volume and Pleasure Responses in the Back Squat Exercise Following a Bench Press.

Rodrigues GM, Machado S, Faria Vieira LA, Ramalho de Oliveira BR, Jesus Abreu MA, Marquez G, Maranhão Neto GA, Lattari E.

J Strength Cond Res. 2021 May 5. doi: 10.1519/JSC.0000000000004054. Online ahead of print.

Transcranial Direct Current Stimulation Does Not Affect Sprint Performance or the Horizontal Force-Velocity Profile.

Alix-Fages C, Garcia-Ramos A, Romero-Arenas S, Nadal GC, Jerez-Martínez A, Colomer-Poveda D, Márquez G.

Res Q Exerc Sport. 2021 Nov 4:1-9. doi: 10.1080/02701367.2021.1893260. Online ahead of print.

Methods

There are some concerns regarding some questions of the methods and design:

Why authors have chosen a single blind approach (only subjects blinded to condition)? Double blind is needed in this type of research because the authors´ a priory expectation could influence the results of the study (e.g.: via changes in the feedback provided, etc).

For sample size estimation authors have chosen the effect size based on previous literature (Hazime et al. (2017) and Lattari et al. (2020a)) who found very large effect sizes. However, all the meta-analysis that studied the effects of a-TDCS on motor performance (strength and endurance) found much lower effect sizes (SMD: 0.20 to 0.40). Please justify this decision, and how it would influence the selection of lower ES.

Regarding the stimulation site, it has been located at the C3/C4 according to the 10-20 electrode placement system. However, leg representation is located in Cz (just in the sulcus), so it is rather difficult to argue that it is possible to focally stimulate left leg M1 using a 5 x 5 cm anode placed over C3. If we take this into account, the “a priory hypothesis” of this study could not be tested.

Line 207 -211: Why authors use different protocols before and after the application of tDCS (anodal/sham)? Subjects warmed-up in the PRE but not after 10 minutes of rest (while applying tDCS). Furthermore, they performed 2 to 3 MVCs in the PRE, but only one in the POST. This could influence the results obtained.

Line 222: please revise the Fatigue Index equation (and the obtained results), it is wrong. It should be as follow:

FI= ((a-b)/a)*100, where “a” is the initial performance and “b” the final performance.

Discussion

Discussion needs substantial improvement. Overall arguments and explanations exposed by the authors are rather superficial and imprecise. As with the rest of the manuscript, writing needs a thoughtful revision. See some specific comments below:

Line 303: fatigability

Line 308-3318: I do not understand why it is hypothesized an increase in the MVC while most of the papers who tested 1RM or MVC failed to demonstrate an effect of tDCS on maximal force production, regardless of the muscle tested (please see Alix-Fages et al., 2019). Kristiansen et al., 2021 previously found a lack of increase in MEP amplitude and MVC in a KE MVC after a-tDCS. It should be better addressed and discussed.

Line 334-337: This is mere speculation. First, M1 excitability has not been measured. Second, it is not well stablished a causal link between M1 increased excitability and force enhancement. Furthermore, this is not aligned with the main interpretation and conclusion of the paper, which is completely opposed to this result.

Line 352-354: This is important, because the task could influence results. See Kaushalya et al., who showed that only TTE test are affected by TDCS during running and cycling endurance activities. Furthermore, recent papers found that aTDCS does not influence performance during repeated “all out” efforts (4-6 sec) interpesed with 30 sec rests (i.e.: RSA test: Alix-Fages et al., 2021). Moreover, as previously commented, the task used in this study only produced a 12% reduction in force overall. This should be discussed further.

Acute effects of transcranial direct current stimulation on cycling and running performance. A systematic review and meta-analysis.

Shyamali Kaushalya F, Romero-Arenas S, García-Ramos A, Colomer-Poveda D, Marquez G.

Eur J Sport Sci. 2021 Jan 7:1-13. doi: 10.1080/17461391.2020.1856933. Online ahead of print

Transcranial direct current stimulation and repeated sprint ability: No effect on sprint performance or ratings of perceived exertion.

Alix-Fages C, Romero-Arenas S, Calderón-Nadal G, Jerez-Martínez A, Pareja-Blanco F, Colomer-Poveda D, Márquez G, Garcia-Ramos A.

Eur J Sport Sci. 2021 Feb 25:1-10. doi: 10.1080/17461391.2021.1883124

Conclusions

“This study found that 10 minutes of 2 mA of a-tDCS is not an effective or consistent method for increasing maximal force production or reducing fatigue in the KE either contralateral or ipsilateral to the stimulated M1.”

This conclusion contrast with those mentioned in lines 334-337.

Reviewer #3: This reviewer is thankful for the opportunity to review this submission by Savoury et al. The aim of the study was to measure the effect of anodal transcranial direct current stimulation (a-tDCS) of the left motor cortex (M1) on both ipsilateral and contralateral knee extension strength and fatigue using a sham-controlled crossover study design. The authors concluded that, contrary to their initial hypothesis, the use of a-tDCS did not significantly increase strength of muscle contraction with contralateral knee extension and significantly decreased muscle contraction with ipsilateral knee extension. They additionally found no significant effect of a-tDCS on muscle fatigue. This work contributes to a growing collection of publications on a-tDCS, all with inconsistent findings regarding its efficacy on muscle contraction augmentation. While the topic is interesting and has promise to translate into clinical space, its findings add to overall ambiguity within the current science. The work is generally well-written, although there is opportunity to provide more clear and consistent use of right/left and contralateral/ipsilateral as shifting between the two sets of terms leads to an unnecessary cognitive burden for the reader. This work provides findings contrary to many published studies, which may be noteworthy, but the authors do not sufficiently reconcile issues they raise in their introduction regarding inconsistency in published tDCS protocol. Consequently, it is difficult to conclude if their contrary findings are a true finding or a result of poorly controlled method with small sample size. With some modifications this work may be fit for publication.

ABSTRACT

Minor Comments

1. Break into sections: Background, Methods, Results, and Conclusion (See Plos One criteria)

INTRODUCTION

Major Comments

1. The authors spend a paragraph describing variables that contribute to the efficacy of tDCS. Even when they describe previously used methods in muscle force production and fatigue, they highlight heterogeneity (i.e., contralateral M1 vs temporal cortex vs prefrontal cortex). It is not well resolved what approach might be the most efficacious. Based on what is written, there seems to be insufficient knowledge to support methodological choices.

2. Moreover, the introduction is hard to follow, unfocused, and too lengthy. The previous research is presented without flow and how it relates to the current study as well as muscular force specifically. Additionally, much of the details should be reserved for the Discussion Section.

3. Lines 148-150: Sex differences did not seem to be an active area for investigation. Authors may add background for this or exclude this hypothesis, as no meaningful investigation was conducted to actually observe differences and as of now seems out of place.

Minor Comments

1. Line 148: Due TO the lack…

2. Throughout the paper, this reviewer suggests using the wording “targeting the left M1” instead of “of the” as it is more reflective of the administration of tDCS

3. Line 79-81: If this sentence remains in the article, make wording clearer with more detailed outcomes.

4. Issues with sentence structure and clearly presenting the previous literature to introduce the topic to the reader.

METHODS

Major Comments

1. Although the authors refer to Thair et al., 2017 paper for screening, it is important to spell out for the reader in the methods the specific Inclusion/exclusion criteria used to enroll participant.

2. Works cited in the methods session regarding intervention design are not addressed in the introduction. Why were particular protocols chosen as opposed to others?

3. The introduction states that this work builds upon the work of Vargas et al. (2018), but this study chose to stimulate and then test versus test at 13 minutes in a 20-mintue stimulation. It is unclear why the authors strayed from this established protocol.

4. Cogiamanian et al. (2007), Kan et al. (2013), Abdelmoula et al. (2016), and Lampropoulou & Nowicky (2013) all use tDCS to measure changes in elbow flexion. The knee and elbow are distinct areas of the motor strip with leg and knee sitting more midline/parasagittal. Montenegro et al. (2015) measured knee flexor, but this was a negative study. Explanation for why this protocol is applicable for your knee flexion study would be helpful.

5. It was noted that measurements of the knee joint were acquired as knee angle can affect the isometric MVC, however the angles were not documented.

6. It is understandable that post-tDCS, participants performed only one MVC to minimize the effect on fatigue protocol, but this leads to a comparison of pre-tDCS measurements after 2 -3 attempts that took participant related effort (change of +5%) into account.

7. Tables mentioned EMG MVC data but were not explicitly stated in the paper; only a strain gauge.

8. Typically blinding of tDCS conditions are implemented in protocols. It does not appear the testing was double-blinded. If so, authors will need to comment on this limitation.

Minor Comments

1. Experimental design noted to be crossover in abstract. Similar verbiage not used in the body of the manuscript. The authors do say, “repeated measures design, with all participants completing four protocols,” which is the equivalent, but there may be some value to maintain consistency.

2. Line 183: 10–20 electroencephalography (EEG) electrode placement system.

3. While testing order was randomized, given the sample size was small, did randomization protocol actually result in equal variances?

4. The Supplementary questionnaire could be improved by asking the participants if they believe they just received the a-tDCS or s-tDCS, but again blinding should be made clearer.

5. Authors should consider reviewing sentence structure, as run-on and missing words make the narrative hard to follow.

RESULTS

Major Comments

1. It would be helpful to report the specific main effects that were not significant

2. Line 282: The title should be more specific. Also, the two paragraphs should be more concise. Moreover, this reviewer does not think this section adds much to the overall aim of the study.

3. Tables 1-3: Discrepancy with Tables and reporting. Table 3 is referenced in-text (Line 290), but as Table 1 in Line 300. Tables 1 and 2 (at the end of the manuscript) are not referenced in-text at all and does not have a Table legend.

4. Figures 2-4: Authors should add more details of the figures to captions

Minor Comments

5. Define a very good and excellent reliability score outside of parentheses.

6. Line 262: the results are presented as ipsilateral with a-tDCS then s-tDCS followed by contralateral with s-tDCS then a-tDCS. Would recommend reversing the order of s-tDCS and a-tDCS in the contralateral group so that the ipsilateral and contralateral are presented using a similar convention.

DISCUSSION

Major Comments

1. First paragraph, it is not commonplace to re-report statistics (i.e., the p-values) in the discussion section. This reviewer suggests removing.

2. First paragraph should include specific details of the results. For example, it appears the authors are including both results of maximal force production and muscle fatiguability in “force impairment.” Please rectify.

3. Lines 320-321: “a-tDCS is not a consistently effective ergogenic aid when the goal is to increase maximal KE force for a discrete contraction”. Is the variation in outcomes (specifically an increase in maximal KE force) a product of the technology or a product of experimental designs with small sample sizes and numerable uncontrolled, confounding variables? (as stated in the next sentence).

4. This reviewer is not sold on the importance of Lines 365-380. These statements can be concise enough to mention in the limitations section (which needs more exploration) due to the fact the participants were not directly asked if they believe they received a-tDCS or s-tDCS.

5. Limitation section needs to be developed. Potentially the comments provided in this review will trigger additional limitations of the study besides the small sample size (e.g., blinding efficacy/expectancy beliefs, other regions of the brain that the authors mention). Other considerations would be the 10-min stimulation, could it be that the length of stimulation inadequate to produce a change?

6. In the conclusion section, the authors mention athletes and performance enhancements, but this was not previously mentioned as a potential impetus of this study. This should be rectified.

7. Lines 392-392 offers a limitation to only studying quadriceps/KE. This should be moved up to the limitations section. Again, same comments for Lines 392-397 are recommendations to limitations and should be discussed in the Limitations section.

Minor Comments

1. Restatement of hypothesis on line 308 is hard to follow. Consider rewording to say, “the hypothesis that …” or adding another hyphen in “a-tDCS-induced..”

2. Would consider changing work reported on line 317 – MVC is measured not reported.

6. PLOS authors have the option to publish the peer review history of their article (what does this mean?). If published, this will include your full peer review and any attached files.

Reviewer #1: No

Reviewer #2: No

Reviewer #3: No

---

## [Author Response · Author response to Decision Letter 0]

20 Aug 2022

Revisions re: data availability have been made as requested. There are no ethical restrictions on data availability. Data has been uploaded to Dryad web site and the DOI is provided.

---

## [Decision Letter · Decision Letter 1]

4 Oct 2022

PONE-D-22-11905R1Reduced isometric knee extensor force following anodal transcranial direct current stimulation of the ipsilateral motor cortexPLOS ONE

Dear Dr. Behm,

Thank you for submitting your manuscript to PLOS ONE. After careful consideration, we feel that it has merit but does not fully meet PLOS ONE’s publication criteria as it currently stands. Therefore, we invite you to submit a revised version of the manuscript that addresses the points raised during the review process.

We look forward to receiving your revised manuscript.

Kind regards,

Xin Ye, Ph.D.

Academic Editor

PLOS ONE

Journal Requirements:

Reviewers' comments:

Reviewer's Responses to Questions

**Comments to the Author**

1. If the authors have adequately addressed your comments raised in a previous round of review and you feel that this manuscript is now acceptable for publication, you may indicate that here to bypass the “Comments to the Author” section, enter your conflict of interest statement in the “Confidential to Editor” section, and submit your "Accept" recommendation.

Reviewer #3: (No Response)

2. Is the manuscript technically sound, and do the data support the conclusions?

Reviewer #3: Partly

3. Has the statistical analysis been performed appropriately and rigorously? 

Reviewer #3: I Don't Know

4. Have the authors made all data underlying the findings in their manuscript fully available?

Reviewer #3: Yes

5. Is the manuscript presented in an intelligible fashion and written in standard English?

Reviewer #3: Yes

6. Review Comments to the Author

Reviewer #3: Summary of feedback:

This reviewer appreciates the consideration of previous feedback and revision of the manuscript. The manuscript has improved, however, this reviewer found it difficult and time-consuming to identify what changes the authors made. On any subsequent revisions, the authors should include a “tracked changes” version, so that reviewers can easily follow the revisions. This reviewer still found sentence structure, clarity issues, and punctuation errors. Attention to detail is imperative. Lastly, more detail should be included in the methods section as to the measures used to assess MVC.

ABSTRACT:

Minor Comments

1. As mentioned in this Reviewer’s previous comments, there is inconsistent use of right/left and contralateral/ipsilateral as shifting between the two sets of terms leads to an unnecessary cognitive burden for the reader. (Lines 42-47 and throughout the manuscript)

2. Line 37, was MVC force tested <immediately> following the a-tDCS and s-tDCS? If so, please specify.

3. Line 40 – remove “The main finding of this study..” and simply say “There was a ….”

4. Line 48 – remove the comma after “Although”

5. Line 50 – remove “Hence”

INTRODUCTION:

Major Comments

1. Third paragraph and fourth paragraph describe positive results and negative results of tDCS, respectively. The third paragraph speaks generally about studies that showed increased muscle force and endurance. The fourth paragraph speaks specifically about tDCS applied to the dorsolateral prefrontal cortex/M1. While the conclusion the authors draw starting on line 77 is likely sound, comparing a general pool of studies to those in a particular area may not be comparing the same phenomenon. The authors discuss this in the final line of paragraph four. If the point to be communicated with the reader is that there is likely heterogeneity in results likely as a result of variation in protocols, this point could likely be made more succinctly.

2. Appreciate the addition of discussion around mechanism and selection of tDCS target added from last version

3. Clear stated aim in final paragraph is helpful - would consider adding ‘left’ prior to M1 in this sentence.

4. The inclusion of sex differences, while appreciated, seems to emerge suddenly. Consider providing some additional context.

5. SICI was not introduced previously to line 112.

6. The differences in sex was outlined without including why this should be studied, despite the lack of research.

Minor comments

1. Errors in punctuation: Lines 64, 70, 72

2. Line 74: prefrontal is generally not hyphenated

3. In line 81, the word “reviews” should be removed and state “meta-analyses” instead.

4. Line 99, a-tDCS was already introduced, and should be used.

5. Line 102: It is unclear to this reader what you mean by “acute studies”. Perhaps, “No studies to date have tested the acute effects of a-tDCS on fatigue/endurance…”?

6. Line 124: Neurones -> neurons (twice)

METHODS:

Minor Comments

1. Appreciate the inclusion of the tDCS Questionnaire in methods. While the results may be in supplementary data, using ‘supplementary’ in the title is confusing. Would consider retitling to “tDCS Blinding Questionnaire”

2. Punctuation Errors: Lines 146, 235

3. Either session or protocol should be used in line 162.

4. Line 182: Consider a different word than “deceptive” – “blinding protocol” would be more appropriate

5. Line 193: use of semi-colon between two and four is unclear. Consider using comma “two, four-second…”

RESULTS:

Minor Comments:

1. Line 243-244 for clarity, consider leading the sentence with “Coefficient of variations …”

2. Line 285: Similar concern about removing supplementary from title, and if truly supplemental, direct readers to the supplemental material within the body of the paragraph

3. Define a very good and excellent reliability score outside of parenthesis.

DISCUSSION:

Major Comments:

1. There is opportunity to discuss more about the gender differences and recommendations for future tests based on why gender was included as exploratory in this study. The introduction referred to a lack of literature, however the discussion noted a previous study. Additionally, in the conclusion that there is no difference between males and females is perhaps overstated. Was the experiment actually powered to observe this? Agree that it is worth noting, but that a recommendation for further research with a specific aim on sex differences is warranted.

2. Line 312: you reiterate the point, “this lack of reliability and high variability in the literature may be also related to the great diversity of implemented protocols (e.g., differences in electrode location, size, number, current density, polarity, and stimulation duration) [42].” after reading the introduction and discussion I am still left wondering how your study attempts to address this. Ideally stronger commentary should be provided as to why your method is optimal and why others should continue to apply this method in future studies.

3. Addition of expanded limitations section is appreciated

4. Lines 350-353 should be included in the introduction as well.

5. Final sentence of your last paragraph again noted heterogeneity in protocol - what is your recommendation?</immediately>

7. PLOS authors have the option to publish the peer review history of their article (what does this mean?). If published, this will include your full peer review and any attached files.

Reviewer #3: No

---

## [Author Response · Author response to Decision Letter 1]

31 Oct 2022

Reviewer #3: Summary of feedback:

This reviewer appreciates the consideration of previous feedback and revision of the manuscript. The manuscript has improved, however, this reviewer found it difficult and time-consuming to identify what changes the authors made. On any subsequent revisions, the authors should include a “tracked changes” version, so that reviewers can easily follow the revisions. This reviewer still found sentence structure, clarity issues, and punctuation errors. Attention to detail is imperative. Lastly, more detail should be included in the methods section as to the measures used to assess MVC.

Response: Based on your and the other reviewer comments we hope this revised version will be acceptable.

ABSTRACT:

Minor Comments

1. As mentioned in this Reviewer’s previous comments, there is inconsistent use of right/left and contralateral/ipsilateral as shifting between the two sets of terms leads to an unnecessary cognitive burden for the reader. (Lines 42-47 and throughout the manuscript)

Response: We have now emphasized the use of ipsilateral and contralateral rather than left and right limbs or muscles throughout the manuscript.

2. Line 37, was MVC force tested following the a-tDCS and s-tDCS? If so, please specify.

Response: As stated in the abstract:

“Knee extensor (KE) MVC force was recorded prior to and following the a-tDCS and s-tDCS protocols. Additionally, a repetitive MVC protocol (12 MVCs with work-rest ratio of 5:10-s) was completed following each tDCS protocol.”

3. Line 40 – remove “The main finding of this study..” and simply say “There was a ….”

Response: Done.

4. Line 48 – remove the comma after “Although”

Response: Done.

5. Line 50 – remove “Hence”

Response: Done.

INTRODUCTION:

Major Comments

1. Third paragraph and fourth paragraph describe positive results and negative results of tDCS, respectively. The third paragraph speaks generally about studies that showed increased muscle force and endurance. The fourth paragraph speaks specifically about tDCS applied to the dorsolateral prefrontal cortex/M1. While the conclusion the authors draw starting on line 77 is likely sound, comparing a general pool of studies to those in a particular area may not be comparing the same phenomenon. The authors discuss this in the final line of paragraph four. If the point to be communicated with the reader is that there is likely heterogeneity in results likely as a result of variation in protocols, this point could likely be made more succinctly.

Response: We have added another sentence as suggested to the second last paragraph of the introduction (prior to the objectives and hypothesis paragraph). Based on your suggestion we state: 

“Furthermore, the heterogeneity and diversity of overall (both sexes) muscle strength, endurance, and corticospinal excitability findings is likely a result of variation in protocols.”

2. Appreciate the addition of discussion around mechanism and selection of tDCS target added from last version

Response: Thank you for the supportive comment.

3. Clear stated aim in final paragraph is helpful - would consider adding ‘left’ prior to M1 in this sentence.

Response: Done.

4. The inclusion of sex differences, while appreciated, seems to emerge suddenly. Consider providing some additional context.

Response: We have added the following contextual information to the introduction.

“There may be sex differences to consider with tDCS. A previous study reported that cathodal tDCS effects were greater in magnitude and duration for female participants, suggesting that female participants may experience increased effects of tDCS when compared to males [82]. Similarly, a multivariate meta-regression by Dedonker et al. (2016) revealed that women demonstrated greater magnitude responses to tDCS, which might be attributed to sex differences in the precise cortical anatomical locations, cognitive task strategies, as well as hormonal differences affecting brain stimulation. A review by Chinzara et al. [43] suggested that the participants’ heterogeneity in terms of sex and genetic diversity requires consideration. Furthermore, the heterogeneity and diversity of overall (both sexes) muscle strength, endurance and corticospinal excitability findings is likely a result of variation in protocols.”

5. SICI was not introduced previously to line 112.

Response: SICI was defined in line 106.

6. The differences in sex was outlined without including why this should be studied, despite the lack of research.

Response: We have added the following contextual information to the introduction.

“There may be sex differences to consider with tDCS. A previous study reported that cathodal tDCS effects were greater in magnitude and duration for female participants, suggesting that female participants may experience increased effects of tDCS when compared to males [82]. Similarly, a multivariate meta-regression by Dedonker et al. (2016) revealed that women demonstrated greater magnitude responses to tDCS, which might be attributed to sex differences in the precise cortical anatomical locations, cognitive task strategies, as well as hormonal differences affecting brain stimulation. A review by Chinzara et al. [43] suggested that the participants’ heterogeneity in terms of sex and genetic diversity requires consideration. Furthermore, the heterogeneity and diversity of overall (both sexes) muscle strength, endurance and corticospinal excitability findings is likely a result of variation in protocols.”

Minor comments

1. Errors in punctuation: Lines 64, 70, 72

2. Line 74: prefrontal is generally not hyphenated

3. In line 81, the word “reviews” should be removed and state “meta-analyses” instead.

4. Line 99, a-tDCS was already introduced, and should be used.

5. Line 102: It is unclear to this reader what you mean by “acute studies”. Perhaps, “No studies to date have tested the acute effects of a-tDCS on fatigue/endurance…”?

6. Line 124: Neurones -> neurons (twice)

Response: All minor comments revised as suggested.

METHODS:

Minor Comments

1. Appreciate the inclusion of the tDCS Questionnaire in methods. While the results may be in supplementary data, using ‘supplementary’ in the title is confusing. Would consider retitling to “tDCS Blinding Questionnaire”

2. Punctuation Errors: Lines 146, 235

3. Either session or protocol should be used in line 162.

4. Line 182: Consider a different word than “deceptive” – “blinding protocol” would be more appropriate

5. Line 193: use of semi-colon between two and four is unclear. Consider using comma “two, four-second…”

Response: All minor comments revised as suggested.

RESULTS:

Minor Comments:

1. Line 243-244 for clarity, consider leading the sentence with “Coefficient of variations …”

2. Line 285: Similar concern about removing supplementary from title, and if truly supplemental, direct readers to the supplemental material within the body of the paragraph

3. Define a very good and excellent reliability score outside of parenthesis.

Response: All minor comments revised as suggested.

DISCUSSION:

Major Comments:

1. There is opportunity to discuss more about the gender differences and recommendations for future tests based on why gender was included as exploratory in this study. The introduction referred to a lack of literature, however the discussion noted a previous study. Additionally, in the conclusion that there is no difference between males and females is perhaps overstated. Was the experiment actually powered to observe this? Agree that it is worth noting, but that a recommendation for further research with a specific aim on sex differences is warranted.

Response: We have expanded the discussion paragraph on sex differences as well as adding a sentence in the limitations section regarding a possible lack of statistical power to detect sex differences.

2. Line 312: you reiterate the point, “this lack of reliability and high variability in the literature may be also related to the great diversity of implemented protocols (e.g., differences in electrode location, size, number, current density, polarity, and stimulation duration) [42].” after reading the introduction and discussion I am still left wondering how your study attempts to address this. Ideally stronger commentary should be provided as to why your method is optimal and why others should continue to apply this method in future studies.

Response: Our methods were based on the prior review recommendations of Savoury et al. Based on the variability in the literature and inconsistent results in the present study there may not be an overall “optimal” method. As discussed in greater detail in the discussion in this revised version, there are sex differences in responses as well as interindividual responses. We previously stated:

“This inter-individual variability likely contributed to some of the non-significant results of this study, and in combination with previous research suggests that a-tDCS is not a consistently effective ergogenic aid when the goal is to increase maximal KE force for a discrete contraction.”

We have added the following sentence based on your suggestion:

“These inconsistent results occurred even though the present study implemented the a-tDCS stimulation recommendations of the Savoury et al. [42] review for the methodological variables that were most likely to produce the greatest exercise performance (i.e., muscle strength, endurance) enhancement.”

3. Addition of expanded limitations section is appreciated

Response: We appreciate the supportive comment.

4. Lines 350-353 should be included in the introduction as well.

Response: This information has been moved to the introduction as suggested.

5. Final sentence of your last paragraph again noted heterogeneity in protocol - what is your recommendation?

Response: We have added the following recommendation:

“A series of comprehensive studies are needed to determine optimal stimulation protocols suitable for various populations (e.g., possible differences between sexes, age), specific performance enhancement (e.g., strength, endurance, or cognitive) and more research targeting areas other than the M1 (e.g., temporal cortex and prefrontal cortex).”

---

## [Decision Letter · Decision Letter 2]

22 Nov 2022

PONE-D-22-11905R2Reduced isometric knee extensor force following anodal transcranial direct current stimulation of the ipsilateral motor cortexPLOS ONE

Dear Dr. Behm,

Thank you for submitting your manuscript to PLOS ONE. After careful consideration, we feel that it has merit but does not fully meet PLOS ONE’s publication criteria as it currently stands. Therefore, we invite you to submit a revised version of the manuscript that addresses the points raised during the review process.

We look forward to receiving your revised manuscript.

Kind regards,

Xin Ye, Ph.D.

Academic Editor

PLOS ONE

Journal Requirements:

Reviewers' comments:

Reviewer's Responses to Questions

**Comments to the Author**

1. If the authors have adequately addressed your comments raised in a previous round of review and you feel that this manuscript is now acceptable for publication, you may indicate that here to bypass the “Comments to the Author” section, enter your conflict of interest statement in the “Confidential to Editor” section, and submit your "Accept" recommendation.

Reviewer #3: All comments have been addressed

2. Is the manuscript technically sound, and do the data support the conclusions?

Reviewer #3: Partly

3. Has the statistical analysis been performed appropriately and rigorously? 

Reviewer #3: I Don't Know

4. Have the authors made all data underlying the findings in their manuscript fully available?

Reviewer #3: Yes

5. Is the manuscript presented in an intelligible fashion and written in standard English?

Reviewer #3: No

6. Review Comments to the Author

Reviewer #3: Summary of feedback:

This reviewer thanks the author for addressing most of the previous comments. This reviewer, however, still identified issues in organization, sentence structure, clarity, and references, especially in the Introduction. This reviewer suggest that the authors recruit a few other researchers, outside of the authors, to review and provide feedback. I offer the additional suggestions, although not exhaustive, below.

ABSTRACT:

Minor Comments

1. Methods: Label the fatigue protocol.

2. Line 44: Replace “impairments” with “reduction” to specify direction of the change.

INTRODUCTION:

Major Comments

1. The Introduction is too long for the information provided. This reviewer suggests being more succinct in reporting information. For example, Lines 82-98 has unnecessary and repetitive details in which the information can be reduced to about 2 sentences. The authors can reserve some of the details for the discussion.

2. Lines 58-62 should be better linked. For example, use of “however” or some other revision to which it would tell the reader, this is what is known clinically without much discrepancy in the literature, however, when investigating the effects of tDCS on exercise and sport performance there appears to be mixed results. This would be a better transition in

3. There are over 60 references in the Introduction alone, which leads to an indication that the information in the introduction is unfocused. The authors should focus in on only articles specific to their research study. For example, are the effects of tDCS on elbow flexion necessary when the current study focuses on lower limb force and endurance? Are studies that target the dlPFC necessary to compare to a study targeting M1?

Minor Comments

1. Lines 139-143: use active voice or make line 139-140 its stand-alone sentence after the stated aim.

METHODS:

Minor Comments

1. Lines 185-187: It is unclear why the authors are using 5 study references just say that M1 is located at C3/C4 on EEG placement. Are the authors trying to say something else? If not, this reviewer thinks 1 major reference of an EEG placement paper is sufficient.

2. Line 194: Remove double word.

RESULTS:

Minor Comments:

1. It would be helpful to restate the statistical models for each analysis in the results since the variables of the ANOVA vary.

DISCUSSION:

Minor Comments:

1. Line 336-338: Make the sentence more detailed based on the references provided (i.e., #10 and #19). For example, the target of tDCS and MVC, etc in these articles.

2. Line 364: Make the effects on sex more specific to the current studies protocol, parameters and variables and not so general.

3. Lines 365-369: Authors should highlight the a-tDCS results first, since it relevant to the current study and then introduce that cathode tDCS generated different results.

4. Line 382: The authors introduced TMS without defining it or providing insight into how TMS may determine potential changes in M1 excitability.

7. PLOS authors have the option to publish the peer review history of their article (what does this mean?). If published, this will include your full peer review and any attached files.

Reviewer #3: No

---

## [Author Response · Author response to Decision Letter 2]

1 Dec 2022

Reviewer #3: Summary of feedback:

This reviewer thanks the author for addressing most of the previous comments. This reviewer, however, still identified issues in organization, sentence structure, clarity, and references, especially in the Introduction. This reviewer suggest that the authors recruit a few other researchers, outside of the authors, to review and provide feedback. I offer the additional suggestions, although not exhaustive, below.

Authors’ response: We thank the reviewer for their contributions to improving the manuscript.

ABSTRACT:

Minor Comments

1. Methods: Label the fatigue protocol.

Authors’ response: We have added the descriptor “fatiguing” to clarify that the repetitive MVC protocol was a fatigue protocol. The revision is as follows:

“Additionally, a repetitive MVC fatiguing protocol (12 MVCs with work-rest ratio of 5:10-s) was completed following each tDCS protocol.” 

2. Line 44: Replace “impairments” with “reduction” to specify direction of the change.

Authors’ response: The term impairment is defined as “diminishment or loss of function or ability.” Hence it does specify the direction of the change. Hence a KE MVC force impairment would be defined as a diminishment or decrease in MVC force function or ability. But if the reviewer likes the word “reduction” better then we have made that change as requested.

INTRODUCTION:

Major Comments

1. The Introduction is too long for the information provided. This reviewer suggests being more succinct in reporting information. For example, Lines 82-98 has unnecessary and repetitive details in which the information can be reduced to about 2 sentences. The authors can reserve some of the details for the discussion.

Authors’ response: We have integrated the information in the suggested paragraph to just 4 sentences. We have also reduced the introduction from about 4 to 3 pages.

2. Lines 58-62 should be better linked. For example, use of “however” or some other revision to which it would tell the reader, this is what is known clinically without much discrepancy in the literature, however, when investigating the effects of tDCS on exercise and sport performance there appears to be mixed results. This would be a better transition in

Authors’ response: We have attempted to better link the information as follows:

“The effectiveness of tDCS for clinical use has shown positive results involving the treatment of depression, anxiety, schizophrenia, Parkinson’s disease, chronic pain, stroke, and other neural-related problems [2-7]. However, the tDCS research on athletic performance is not as consistent.”

3. There are over 60 references in the Introduction alone, which leads to an indication that the information in the introduction is unfocused. The authors should focus in on only articles specific to their research study. For example, are the effects of tDCS on elbow flexion necessary when the current study focuses on lower limb force and endurance? Are studies that target the dlPFC necessary to compare to a study targeting M1?

Authors’ response: Nineteen (19) of the citations are provided for one statement which demonstrated:

“…that tDCS is effective at increasing maximal muscle force and endurance [8-25] as well as strength training volume [26].”

Another two sentences illustrate the expansiveness of the non-significant literature with 15 citations:

“…However, many others report no significant effects or decreases of muscle force or endurance when stimulating the motor cortex [9,21,28-37]. There were also no significant effects on jump height [38], or single or repeated sprint performance [39,40] when stimulating the dorsolateral prefrontal cortex.”

Hence, 34 of the citations are focussed on either the performance improvements or the reductions associated with tDCS. We feel it is necessary to comprehensively cover the literature so the reader is familiar with the disparity and conflicts in the prior publications and what has been examined before.

Based on the reviewer’s comment we have reduced the introduction by approximately a page.

Minor Comments

1. Lines 139-143: use active voice or make line 139-140 its stand-alone sentence after the stated aim.

Authors’ response: We have removed the first part of the sentence regarding the type of stimulation protocol. We have moved this information to the methods section.

METHODS:

Minor Comments

1. Lines 185-187: It is unclear why the authors are using 5 study references just say that M1 is located at C3/C4 on EEG placement. Are the authors trying to say something else? If not, this reviewer thinks 1 major reference of an EEG placement paper is sufficient.

Authors’ response: We would like to illustrate to the reader that this stimulation protocol has been widely used. Thus, the reader should now have confidence that this protocol is valid and reliable.

2. Line 194: Remove double word.

Authors’ response: The second repeated word (i.e., protocol) was removed as suggested.

RESULTS:

Minor Comments:

1. It would be helpful to restate the statistical models for each analysis in the results since the variables of the ANOVA vary.

Authors’ response: A 3-way repeated measures ANOVA was employed for all testing.

1. Absolute MVC force

2. Relative MVC force

3. Fatigue test.

Hence stating that it was a 3-way repeated measures ANOVA for each measure would be redundant or repetitive with the information already provided in statistical analysis section and would not provide any greater clarity.

With each interaction in the results, we state the variables tested. Below are just 3 examples:

“…with no significant interaction effects found for condition (condition x time), or leg tested (leg tested x time), nor main effects for condition.” 

“A significant [F(1,60) =7.156, p = 0.010, ηp2 = 0.11] interaction effect was found for condition and leg tested (condition x leg tested x time)(Fig 2).”

“A significant interaction effect for condition x leg tested [F(1,56) = 8.12, p = 0.006, ηp2 = 0.13], showed…”

DISCUSSION:

Minor Comments:

1. Line 336-338: Make the sentence more detailed based on the references provided (i.e., #10 and #19). For example, the target of tDCS and MVC, etc in these articles.

Authors’ response: We have added the following details as suggested.

“However, a previous studies involving a-tDCS of the contralateral leg motor cortex reported improved foot pinch force for 30 minutes post-a-tDCS [10], as well as increased KE MVC force for 60 minutes after a-tDCS [19].”

2. Line 364: Make the effects on sex more specific to the current studies protocol, parameters and variables and not so general.

Authors’ response: Line 364 is the last sentence of the previous paragraph and does not discuss sex differences. However, the following paragraph does discuss sex differences and we have added some greater detail. For example: we have added the following information to that paragraph.

A review by Dedoncker [62] suggested that women demonstrated greater accuracy on cognitive tasks with a-tDCS. A previous study that retrospectively re-analyzed data collected from previous transcranial direct current stimulation studies did not report significant cortical excitability differences between the sexes for a-tDCS. However, the effects of cathodal tDCS on neuroplasticity were greater, lasted longer and demonstrated more inhibition for female participants in comparison to males, suggesting that female participants may experience increased effects of tDCS possibly due to the effects of sex hormones [61].

3. Lines 365-369: Authors should highlight the a-tDCS results first, since it relevant to the current study and then introduce that cathode tDCS generated different results.

Authors’ response: We have changed the order as suggested.

4. Line 382: The authors introduced TMS without defining it or providing insight into how TMS may determine potential changes in M1 excitability.

Authors’ response: We have added the following explanation:

“Furthermore, as only the KE (quadriceps) were tested, future studies should aim to determine if other muscle groups ipsilateral to the site of stimulation are affected in a similar manner, while also utilizing transcranial magnetic stimulation (TMS) to determine potential changes in corticospinal excitability. When combined with transmastoid electrical stimulation, which activates axons in the spinal cord, effects can be distinguished between cortical and spinal excitability or inhibition.”

---

## [Decision Letter · Decision Letter 3]

21 Dec 2022

Reduced isometric knee extensor force following anodal transcranial direct current stimulation of the ipsilateral motor cortex

PONE-D-22-11905R3

Dear Dr. Behm,

We’re pleased to inform you that your manuscript has been judged scientifically suitable for publication and will be formally accepted for publication once it meets all outstanding technical requirements.

Kind regards,

Xin Ye, Ph.D.

Academic Editor

PLOS ONE

Additional Editor Comments (optional):

Reviewers' comments:

Reviewer's Responses to Questions

**Comments to the Author**

1. If the authors have adequately addressed your comments raised in a previous round of review and you feel that this manuscript is now acceptable for publication, you may indicate that here to bypass the “Comments to the Author” section, enter your conflict of interest statement in the “Confidential to Editor” section, and submit your "Accept" recommendation.

Reviewer #3: (No Response)

2. Is the manuscript technically sound, and do the data support the conclusions?

Reviewer #3: Yes

3. Has the statistical analysis been performed appropriately and rigorously? 

Reviewer #3: I Don't Know

4. Have the authors made all data underlying the findings in their manuscript fully available?

Reviewer #3: Yes

5. Is the manuscript presented in an intelligible fashion and written in standard English?

Reviewer #3: Yes

6. Review Comments to the Author

Reviewer #3: This Reviewer thinks that the manuscript has improved greatly since initial review. This Reviewer does not have any other MAJOR comments, and just a few MINOR comments. Once those are fixed, this Reviewer believes it has reached a point for acceptance.

1. Page 5 Line 105 AND Page 15 Line 363: The Page 5 can be used to define TMS and Page 15 can be moved to abbreviation.

2. Page 7 Line 161: Revise for clarity. This sentence is missing a word or two.

3. Page 9 Line 201-101 Revise for clarity.

4. Page 13 Line 35: Revise to "....may also be related to greater diversity..."

5. Page 15 Line 346: Authors can use the abbreviation tDCS here.

7. PLOS authors have the option to publish the peer review history of their article (what does this mean?). If published, this will include your full peer review and any attached files.

Reviewer #3: No

---

## [Editor Report · Acceptance letter]

27 Dec 2022

PONE-D-22-11905R3 

Reduced isometric knee extensor force following anodal transcranial direct current stimulation of the ipsilateral motor cortex 

Dear Dr. Behm:

I'm pleased to inform you that your manuscript has been deemed suitable for publication in PLOS ONE. Congratulations! Your manuscript is now with our production department. 

Kind regards, 

on behalf of

Dr. Xin Ye 

Academic Editor

PLOS ONE